# When RNA goes off script: ensuring transcript fidelity in transgene expression

Rachel Anderson [ID] [1,2], Christalyn Ausler [ID] [1] & Ankur Jain [ID] [1,2] [✉]

## Abstract

**Plasmids are the workhorses of molecular biology: fast, flexible, and often taken for granted. We clone, overexpress, tag, and mutate freely, assuming they will faithfully produce RNA transcripts that match the intended DNA sequence. This assumption is rarely tested and often invalidated. Sequences in plasmid backbones, epitope tags, and codon-optimized regions may inadvertently harbor cryptic promoters or splice sites. The resulting unexpected transcripts and proteins, while often undetected, can distort results and propagate false conclusions through papers, grants, and even clinical trials. In this perspective, we highlight published cases where plasmids have distorted results and misled interpretation. We examine the mechanisms and consequences of plasmid-associated expression artifacts and offer practical strategies to minimize them. Finally, we call for a revision of community standards for experiments using transgenes: deposit complete plasmid sequences and verify the resulting transcripts using RNA-seq.**

**Keywords** Plasmids; Cryptic Promoters; Aberrant Splicing; Codon Optimization; RNA Processing

## Introduction

The speed, efficiency, and affordability of transgene expression via recombinant plasmids have enabled researchers to express nearly any gene on demand across a wide range of cell types and organisms. This accessibility has revolutionized biomedical research, facilitating detailed explorations into gene function, production of therapeutic proteins such as insulin, and the development of gene therapies (Prazeres and Monteiro, 2014). Modern expression vectors are true marvels of genetic engineering that integrate regulatory elements adapted from many organisms to achieve robust transgene expression and streamline experimental workflows (Williams et al, 2009; Nora et al, 2019). For instance, plasmids commonly feature bacterial origins of replication and antibiotic selection markers to enable efficient propagation in *E. coli* (Williams et al, 2009; Nora et al, 2019). Transgenes are cloned under strong constitutive or inducible promoters, often derived from viral sources (Williams et al, 2009; Nora et al, 2019). To enhance RNA processing, stability, and export, these vectors incorporate post-transcriptional regulatory sequences such as polyadenylation signals, synthetic introns, and viral-derived elements like the woodchuck hepatitis virus posttranscriptional regulatory element (WPRE) (Williams et al, 2009; Nora et al, 2019). Additionally, multiple cloning sites composed of numerous restriction enzyme recognition sequences enable flexible and convenient gene insertion (Williams et al, 2009; Nora et al, 2019). Together, these modular features have made plasmids highly adaptable "plug-and-play" platforms that form the cornerstone of modern molecular biology.

However, it is crucial to recognize that non-native sequences in these vectors are not always benign and can exert unintended functions, which we will focus on in this perspective. A common assumption in experiments involving transgenes is that inserting a minimal cDNA into a plasmid will yield the desired protein product. While this is a reasonable assumption in prokaryotes, where nascent RNA is co-transcriptionally translated, it does not necessarily hold true in eukaryotes, where transcription and translation are physically separated. In higher eukaryotes, nearly all transcripts are extensively modified before encountering the translational machinery. Most human genes contain multiple introns, which can comprise over 90% of the gene's length (Lee and Rio, 2015). Alternative splicing of these introns expands protein diversity but is also a major source of error, potentially leading to non-functional or deleterious protein isoforms (Daguenet et al, 2015). Likewise, transcription termination, where the 3' end of nascent mRNAs is cleaved and polyadenylated, is a key step in mRNA maturation. The use of alternative polyadenylation sites can generate isoforms with different stability, localization, or coding potential (Di Giammartino et al, 2011). In addition, base modifications can influence splicing, degradation, translation efficiency, and even alter the encoded protein sequence (Lewis et al, 2017).

Given the complexity of RNA processing in eukaryotic cells, it is unsurprising that exogenously expressed transgenes are also subject to processing by the host cells' machinery. The signals guiding these post-transcriptional steps are short and degenerate, often encoded directly within the transcript. Unintended recognition of foreign elements that resemble endogenous regulatory motifs can generate mature transcripts that diverge from the intended design. While most native mRNAs are present at 10–100 copies per cell, and even the most abundant rarely exceed ~1000 copies (Marinov et al, 2014), transgenes driven by strong viral promoters can reach expression levels of 1000–10,000 copies per

[1]Whitehead Institute for Biomedical Research, 455 Main Street, Cambridge, MA 02142, USA. [2]Department of Biology, Massachusetts Institute of Technology, 31 Ames Street, Cambridge, MA 02139, USA. [✉]E-mail: ajain@wi.mit.edu

https://doi.org/10.1038/s44318-026-00733-z | Published online: 16 March 2026

cell (Zhou et al, 2023; Ottesen et al, 2024). At this level of expression, even rare misprocessed transcripts can rival or exceed the abundance of many endogenous mRNAs and profoundly influence cell fate.

In several notable cases, artifacts arising from transgene systems have misled scientists to erroneous conclusions about underlying biological mechanisms that were later proven incorrect when the endogenous genes were examined. For example, apparent ribosomal frameshifting in *CCR5* (Khan et al, 2022) and *ATP7B* (Loughran et al, 2022), noncanonical translation initiation of *XIAP* (Lemp et al, 2012), discrepancies among promoter-enhancer assays (Muerdter et al, 2018), novel viral protein isoforms (Majerciak and Zheng, 2016), and interference between co-transfected plasmids (Nejepinska et al, 2012) were all ultimately traced to unintended RNA processing in synthetic constructs. Despite these cautionary examples, it remains common practice to assume, often without rigorous validation, that transgenes faithfully and exclusively produce the intended cDNA sequence in mature mRNA. This assumption can lead to artifacts that distort mechanistic interpretations and can take years, or even decades, for the scientific community to fully unravel, resulting in wasted time and resources.

In this perspective, we highlight how aberrant transcription and splicing can undermine the fidelity of transgene expression in mammalian cells. We draw on specific examples where widely used vectors have led to RNA and protein products that deviate from the intended sequence with potential to produce confounding results. We also provide guidelines when designing experiments with exogenous DNA in mammalian cells that can help minimize undesirable RNA processing, as well as suggest essential controls to avoid common pitfalls. While not exhaustive, our goal is to raise awareness about these known issues, offer actionable design and validation strategies, and encourage a more critical approach to interpreting data from plasmid-based systems. We believe that a broader recognition of these issues will drive the development of more robust and predictable vectors, ultimately improving the reproducibility and reliability of these foundational tools.

## Improvised lines: when unintended promoters take the stage

Plasmids used for transgene expression in mammalian cells typically contain a eukaryotic promoter immediately upstream of the gene of interest, which is assumed to be the sole driver of transcription (Fig. 1A, and expected outcome Fig. 1C). However, unintended promoter activity from the plasmid backbone or within the gene can generate aberrant transcripts that can confound experimental interpretation. This problem is particularly acute with plasmid-based episomal expression systems, where transgenes are uncoupled from endogenous chromatin regulation that would normally suppresses cryptic promoter activity (Jensen et al, 2013).

Spurious promoter activity may sometimes be benign, resulting in transcripts that are not translated or are rapidly degraded. However, transgenes are often deliberately overexpressed, delivered at high plasmid copy number, or through multi-copy viral integrations. In such cases, even rare transcriptional events can produce aberrant RNAs at levels comparable to, or exceeding those of median endogenous genes (~10 copies per cell) (Marinov et al, 2014). These aberrant transcripts may encode unintended proteins (Lemp et al, 2012), generate noncoding RNAs that can disrupt regulatory networks (Mao et al, 2025), or trigger cellular stress responses (Nejepinska et al, 2014), which can substantially alter cellular behavior. Notably, these aberrant transcripts may still be transgene-specific. They may contain portions of the transgene, or the promoter itself may reside within the transgene sequence. This creates a significant interpretive challenge: phenotypic effects observed when comparing control and transgene-expressing constructs are typically attributed to the intended full-length protein, but may instead arise from unintended RNA species produced by aberrant transcription. In this section, we examine how various sources of unintended promoter activity produce sequence-dependent effects at both RNA and protein levels.

### Surprise entrances: cryptic promoters within transgenes

Perhaps the most insidious source of aberrant transcription lies within the transgene sequence itself. The transgene may harbor promoter-like sequences that can initiate transcription downstream of the intended start site (Lemp et al, 2012; Nejepinska et al, 2012; Akirtava et al, 2022; Vopálenský et al, 2008; Chaudhari et al, 2022) (Fig. 1A). Eukaryotic genomes have many cryptic transcription initiation sites that are normally silenced by chromatin-associated regulatory mechanisms (Hennig and Fischer, 2013). However, plasmid-expressed transgenes and their surrounding synthetic sequences may lack critical regulators, leading to re-activation of these initiation sites. Such internal promoters can generate truncated mRNA isoforms that bypass regulatory elements in the 5' UTR (untranslated region) or even the intended start codon. These truncated RNAs may expose downstream AUG codons, leading to reduced full-length protein expression or production of alternative isoforms (Fig. 1C) that may retain partial function, interfere with the full-length protein, or gain unintended activities. Notably, many human genes naturally harbor additional in-frame AUG codons downstream from the annotated translation start site; initiation at these sites allows regulated expression of alternative isoforms (e.g., by removing localization signals), that can compete with or modulate the function of the full-length protein (Van Damme et al, 2014; Benitez-Cantos et al, 2020; Ly et al, 2025).

Alternatively, if the retained AUG start codons are in a different reading frame, these RNAs may encode frameshifted proteins with unpredictable activities (Fig. 1C). Translation of these alternative frameshifted ORFs is difficult to detect and not commonly assayed. However, in several cases, investigators have sought, and found, products of alternative frameshifted ORFs in plasmid-expressed genes (Schirmbeck et al, 2005; Delcourt et al, 2018). For example, in DNA vaccines, such cryptic peptides have been shown to elicit CD8[+] T cell responses against unintended antigens (Schirmbeck et al, 2005). Internal promoters may also reside in commonly used tags and reporters. For instance, the widely used firefly luciferase, a staple in gene expression assays, harbors a cryptic internal promoter capable of initiating transcription in both yeast and mammalian cells (Vopálenský et al, 2008). More broadly, in fusion constructs, internal initiation events can give rise to unexpected protein products in alternative reading frames or even generate reporter-positive fragments that decouple the reporter signal from the full-length transgene, thus confounding interpretation.

A particularly striking consequence of internal promoter activity has been the

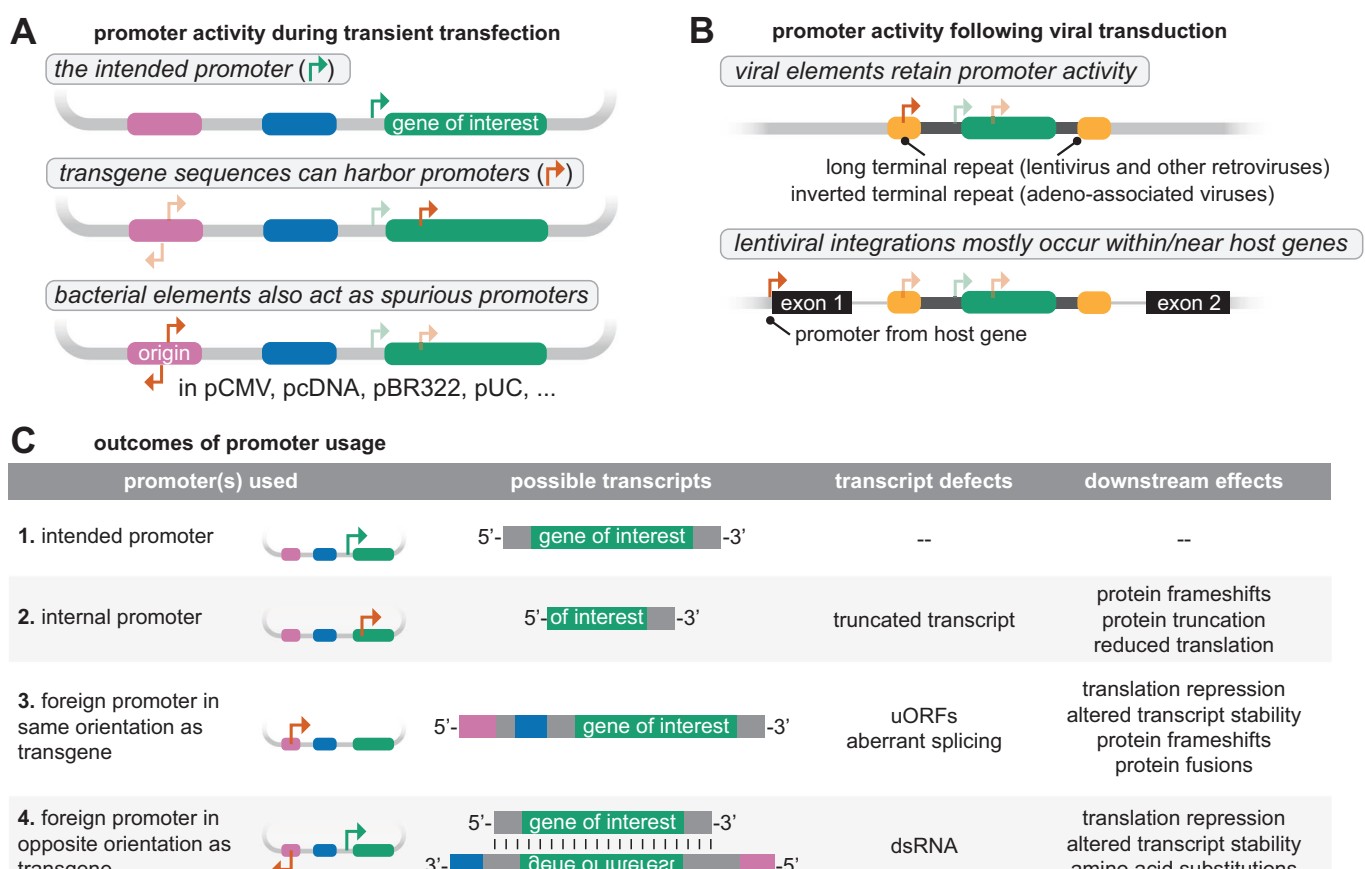

**Figure 1. Sources and consequences of spurious promoter activity.**

(A) Sources of spurious promoter activity in transiently expressed plasmids. Putative promoters are indicated by right-facing arrows. Transcription is typically expected to initiate from the promoter immediately upstream of the transgene, but transgenes and their regulatory regions, such as 5′ UTRs, may also contain cryptic promoters. Cryptic promoter activity can also arise from plasmid elements such as the ColE1-family origins of replication, which are ubiquitous in modern plasmids (e.g., in existing mammalian expression systems like pCMV, pcDNA, pDest26, pHR, and their derivatives, as well as in bacterial expression and cloning plasmids such as pBluescript, pUC, and pBR322 when re-purposed for use in mammalian systems)(Trivedi et al, 2014). (B) Sources of unintended promoter activity in stably expressed transgenes. Viral elements such as long terminal repeats (LTRs) and inverted terminal repeats (ITRs) retain promoter activity and can initiate transcription. A lentiviral cassette is shown integrated within a host intron, with host exons depicted in black. Lentiviral vectors frequently integrate near active genes, and the transgene may be transcribed as part of a chimeric RNA driven by a proximal host promoter. (C) Diverse RNA products can arise from unintended promoter activity. (1) Transcription from the expected upstream promoter produces the intended transcript. (2) Internal promoters within the transgene can generate truncated transcripts that bypass upstream regulatory elements, potentially initiating at downstream AUGs and producing shortened or frameshifted proteins. (3) Upstream cryptic promoters within plasmid backbones or host DNA can produce extended 5′ UTRs containing upstream open reading frames (uORFs), splice sites, or other regulatory features that impair translation of the intended ORF. (4) Antisense transcription from reverse-oriented promoters can generate double-stranded RNA, triggering translational repression, degradation, or adenosine-to-inosine RNA editing.

widespread misidentification of internal ribosome entry site (IRES) activity. For example, translation of the hypoxia-inducible factor *HIF1A* that was originally attributed to IRES activity during hypoxic stress was later shown to result from canonical translation of transcripts initiated at a cryptic promoter within the gene body, a phenomenon not reported under physiological expression of the endogenous gene (Liu et al, 2005; Bert et al, 2006). Similar misattributions of IRES activity have been reported for *CDKN1C* (Young et al, 2008), *GLUT1* (Young et al, 2008), *HIF2A* (Young et al, 2008), *HOXA9* (Akirtava et al, 2022), *MYC* (Lemp et al, 2012), *PIM1* (Wang, 2005), *SPAST* (Mancuso and Rugarli, 2008), *TPI* (Young et al, 2008), and *VEGF* (Young et al, 2008), where IRES-like activity was ultimately explained by internally initiated transcripts rather than genuine cap-independent translation. These findings have prompted calls for more rigorous experimental standards, including the use of promoter-less constructs and transcript-specific knockdowns, to accurately distinguish authentic IRES activity from RNA processing artifacts (Jackson, 2013; Terenin et al, 2017; Loughran et al, 2025). Taken together, these examples demonstrate that internal promoter activity is not an isolated anomaly but a common source of spurious RNAs that can influence phenotypes and mislead experimental conclusions.

## Backstage voices: ColE1 and other plasmid elements

Beyond internal promoters within the transgene, the plasmid backbone (often comprising 70–80% of the vector sequence) harbors elements that can also exhibit

cryptic promoter activity (Chauhan et al, 2009). A prominent example is the ColE1 origin of replication, which is commonly used for plasmid propagation in *E. coli*. This sequence coincidentally contains eukaryotic promoter motifs (such as a TATA box and GC box) and acts as a cryptic promoter in eukaryotic cells (Lemp et al, 2012; Muerdter et al, 2018) (Fig. 1A). Upon plasmid transfection in mammalian cells, it can drive expression at levels comparable to those from strong viral promoters, like CMV (cytomegalovirus) and SV40 (simian virus 40) promoters (Lemp et al, 2012; Muerdter et al, 2018; Anderson et al, 2024). In fact, the ColE1 element has even been used as a core promoter in enhancer screens due to its high activity and favorable signal-to-noise characteristics (Muerdter et al, 2018).

When the transgene is oriented in the same direction as ColE1-driven transcription, it may produce read-through transcripts that fuse portions of the vector backbone to the gene of interest (Lemp et al, 2012; Muerdter et al, 2018; Anderson et al, 2024) (Fig. 1A,C). These chimeric RNAs with extended 5' fusions can interfere with transgene expression in multiple ways (Fig. 1C). During translation initiation, the ribosome begins at the 5' end of the transcript and scans for a suitable AUG codon. If the chimeric 5' region includes upstream open reading frames (uORFs), the ribosome may initiate translation there, reducing expression of the intended protein (Calvo et al, 2009) (Fig. 1C). Upstream AUGs that are in-frame with the transgene and lack intervening stop codons can yield N-terminally extended fusion proteins (Anderson et al, 2024; Calvo et al, 2009; Fukunaga et al, 2009) (Fig. 1C). Alternatively, out-of-frame AUGs without intervening stop codons may generate unrelated peptides with unpredictable functions (Fig. 1C). In addition to altering coding capacity, these upstream regions may inadvertently contain RNA regulatory motifs that affect transcript stability, nuclear export, and sub-cellular localization (Lewis et al, 2017). Moreover, these chimeric regions can also introduce splice donor or acceptor sites, giving rise to unanticipated splice variants and unexpected protein products (Fig. 1C), which will be discussed in greater detail later.

Conversely, if the promoter in the ColE1 origin is in the reverse orientation to the transgene, it could drive expression of antisense RNAs that are complementary to the gene of interest. These antisense RNAs can hybridize with the sense transcript to form double-stranded RNA (dsRNA, Fig. 1C) (Chalupnikova et al, 2013). Cells recognize dsRNA as a marker of viral infection and activate various innate immune response mechanisms, including stress response pathways, cytokine production, and global translation inhibition (Chalupnikova et al, 2013; Hur, 2019; Nejepinska et al, 2014). Notably, these effects can occur in *trans* when plasmids are co-transfected (Nejepinska et al, 2014). For example, antisense transcription from a neomycin resistance cassette has been shown to induce dsRNA-mediated translational inhibition of co-transfected luciferase reporters (Nejepinska et al, 2012). Additionally, dsRNAs are substrates for adenosine deaminases acting on RNA (ADARs), which catalyze adenosine (A) to inosine (I) editing (Hur, 2019; Walkley and Li, 2017). Because inosine can base-pair like guanosine, these edits can alter splice-site recognition (Rueter et al, 1999) or recode codons during translation (Walkley and Li, 2017) (Fig. 1C).

## Unwanted encores: terminal repeats and integration artifacts

In contrast to transiently expressed plasmids, lentiviral (and other retroviral) and adeno-associated virus (AAV) vectors are commonly used for stable transgene delivery. Both vector types contain repetitive regions that are essential for their respective viral lifecycles (Fig. 1B): long terminal repeats (LTRs) in lentiviral vectors and inverted terminal repeats (ITRs) in AAV vectors (Bulcha et al, 2021). While lentiviral vectors typically integrate into the host genome through LTR-mediated mechanisms, AAV vectors often persist as extrachromosomal episomes, though they can integrate at defined loci such as AAVS1 under specific conditions (Bulcha et al, 2021). Problematically, in their native roles, both LTRs and ITRs evolved to initiate transcription of full-length viral RNA and coordinate replication or packaging. The residual promoter activities of these repeats can inadvertently drive expression of aberrant transcripts containing portions of the viral cassette and the transgene (Bulcha et al, 2021; Earley et al, 2020) (Fig. 1B,C). In experiments using fluorescent reporters, LTR-produced transcripts in HEK-293 and K562 cells are expressed at levels comparable to those driven by strong viral promoters (Knight et al, 2010; Weber and Cannon, 2007). Similarly, viral ITR promoter activity has been harnessed to express CFTR in a gene therapy trial (Flotte et al, 1996). In most experimental settings, however, such uncontrolled transcription is problematic. Self-inactivating (SIN) lentiviral vectors mitigate this issue by mutating promoter elements in the LTR, but this comes at the cost of reduced viral titers (Zufferey et al, 1998).

Beyond spurious promoter activity, both lentiviral and AAV vectors can place transgenes in unexpected sequence contexts that could alter their expression, sequence, and function (Fig. 1B,C). Lentiviruses preferentially integrate near or within transcriptionally active genes (Moiani et al, 2012; Cesana et al, 2012; Almarza et al, 2011) (Fig. 1B), while non-integrated AAV vectors tend to concatemerize over time (Penaud-Budloo et al, 2008). These events can generate chimeric transcripts that fuse transgene sequences with host or vector-derived regions, placing the transgene in non-native sequence contexts that may expose spurious open reading frames, cryptic splice sites (Knight et al, 2010; Moiani et al, 2012; Almarza et al, 2011; De Ravin et al, 2022), or other unintended regulatory motifs (Moiani et al, 2012; Cesana et al, 2012; Almarza et al, 2011), as discussed above (Fig. 1C). Notably, integration within host genes and formation of fusion transcripts form the basis of gene trapping experiments (Lai et al, 2002). However, in routine expression studies, these events are highly undesirable and can confound the interpretation of transgene behavior.

Stable integration can also be achieved by transfecting linearized DNA, such as during knock-in experiments, but this method introduces its own set of potential artifacts. Insertions can occur at multiple loci across the genome, or as tandem arrays within a single locus. Within these arrays, residual bacterial backbone or truncated fragments of the expression cassette may persist. Individual copies may be inverted, form head-to-tail concatemers, or recombine with other homologous genomic sequences (Nicholls et al, 2019; Luqman et al, 2025). These integration details are rarely characterized, and each of these factors, from copy number to orientation, can result in transcripts that deviate substantially from the intended design.

## Casting the wrong star: clonal selection amplifies artifacts

Experiments with monoclonal populations of stably integrated transgenes are particularly vulnerable to artifacts arising from aberrant insertion events described above. In one estimate, 40% of cells transduced at ~5 viral copies per cell showed host-lentiviral fusion transcripts (Moiani et al, 2012). In polyclonal populations, each cell typically contains distinct viral integration events that are averaged out in a bulk assay (such as a western blot). However, when isolating monoclonal populations, one may inadvertently select a clone where spurious transgene fusions significantly contribute to the phenotype (Knight et al, 2010; Moiani et al, 2012; Almarza et al, 2011; De Ravin et al, 2022). Accordingly, studies using monoclonal lines should incorporate replicate clones and transcript-level validation to avoid misattributing effects to the intended transgene. Polyclonal pools are not entirely refractory to these issues. Transgenic insertions may impart a growth or survival advantage, and during prolonged culture, these rapidly-dividing cells can dominate the population (Modlich et al, 2009). For instance, in lineage tracking experiments performed on K562 cells, a small proportion of clones (~10) made up half of the population after 90 doublings (Porter et al, 2014). Although techniques exist to monitor clonality (e.g., lineage tracking or mapping insertion sites by splinkerette PCR (Uren et al, 2009)), limiting the use of high-passage transgenic cell lines may help reduce the risk of clonal sweeps.

## Stage directions gone awry: the drama of aberrant splicing

In eukaryotes, nascent transcripts undergo extensive processing, including splicing, 3′ end cleavage, polyadenylation, base modification, and nuclear export. Many of these steps are guided by weakly conserved sequence motifs that are interpreted in a context-dependent manner. Unintended interactions of the transgene with the host processing machinery can generate isoforms that deviate from the intended transcript design.

Splicing is a particularly common source of such divergence. The spliceosome identifies and processes transcripts using sequence signals at the 5′ and 3′ splice sites and the branch point (Lee and Rio, 2015) (Fig. 2A). In humans, these core motifs are short and degenerate: the 5′ splice site nearly always contains a 'GU' dinucleotide and the 3′ splice site an 'AG', but the surrounding nucleotides tolerate substantial variations (Lee and Rio, 2015) (Fig. 2A). Cells regulate splice site usage through nearby splicing regulatory sequences, which recruit RNA-binding proteins (RBPs) to the nascent transcript (Lee and Rio, 2015) (Fig. 2A). The motifs recognized by these RBPs are also relatively short, such as the "GCAUG" sequence recognized by RBFOX2 (Tao et al, 2024). As a result, even sequences not intentionally designed for splicing can be erroneously recognized and processed by the spliceosome. Moreover, the regulatory impact of these elements can change depending on its position relative to a splice site. A given sequence may enhance splicing when located upstream of a splice site but repress it when positioned downstream (Lee and Rio, 2015). Thus, even small sequence changes may inadvertently create, destroy, or move RNA processing signals, altering the fate of the transcript and resulting proteins in ways that are often difficult to predict a priori (Fig. 2B).

Transgene sequences derived from cDNA (which are copied from mature, fully processed mRNAs) are generally not expected to undergo further splicing, as they lack introns and are typically depleted of splicing-promoting motifs. Nonetheless, unintended splicing has been widely observed in plasmid-expressed cDNAs derived from many genes, including *ABCB1*

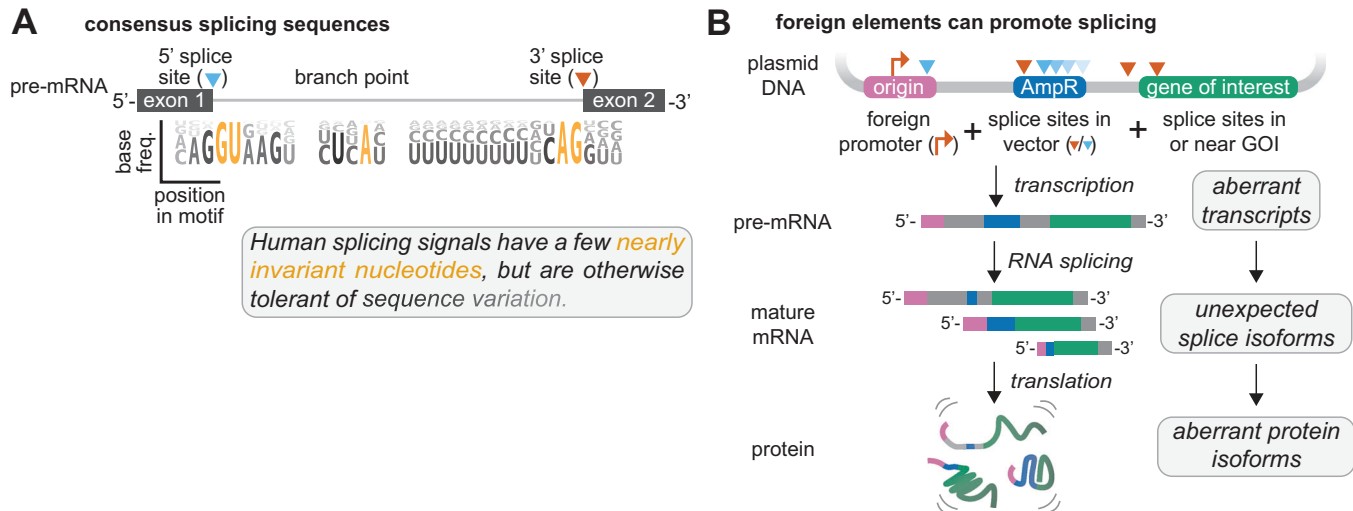

**Figure 2. Transgene and plasmid elements can work in tandem to drive aberrant RNA processing.**

(A) Minimal sequence features involved in canonical splicing. Exons (black boxes) are separated by an intron (gray line). Human consensus motifs for the 5′ and 3′ splice site (triangles) and the branch point are shown as sequence logos (Gao et al, 2008; Ma et al, 2015), highlighting the degenerate nature of splicing signals. (B) Aberrant transcription initiation and splicing from the plasmid backbone can produce mature mRNAs that encode unexpected, out-of-frame proteins. In this example, cryptic transcription from the ColE1 origin and splicing from the AmpR gene into the transgene generate chimeric mRNAs. These transcripts contain ORFs that encode frameshifted portions of the transgene, and the resulting proteins were initially attributed to non-AUG translation, illustrating how unintended RNA processing from vector elements can mislead mechanistic interpretations.

(Sorrentino et al, 1995), *ACE2* (Tomberg et al, 2021), *ALKBH5* (Tomberg et al, 2021), *CAPN10* (Ono et al, 2022), *CCR5* (Khan et al, 2022), *DUX4* (Ansseau et al, 2015), *EIF4G* (Baranick et al, 2008), *ERCC1* (Bosma et al, 2003), *FACI* (and 17 other genes) (Cheng et al, 2022), *HBG1/HBG2* (Zeglinski et al, 2024), *IL1RN* (Lee et al, 2007), *MYC* (Lemp et al, 2012), *NRF1* (Baranick et al, 2008), *RBM3* (Baranick et al, 2008), and *XIAP* (Lemp et al, 2012; Baranick et al, 2008; Saffran and Smiley, 2009). In several cases, seemingly intriguing phenotypes from plasmid-expressed transgenes (such as apparent IRES activity) were ultimately traced to aberrantly spliced transcripts (Fig. 2B) and often do not occur when the endogenous gene is examined.

In other cases, transgenes may be built from cDNA elements with significant synthetic contexts. For instance, massive parallel reporter assays (MPRAs) place a library of candidate sequences into non-native surroundings to test for features like promoter activity, RNA stability, or translation efficiency (through interactions with transcription factors (Arnold et al, 2013) and RNA-binding proteins (Cottrell et al, 2018)). However, these highly synthetic architectures in MPRAs often unintentionally introduce RNA processing signals, confounding the readouts (Muerdter et al, 2018; Dao et al, 2025). As an example, circular RNA expression plasmids use flanking sequences with complementary inverted repeats to drive circularization of the RNA transcript by spliceosome-mediated back-splicing (Liang and Wilusz, 2014). While this system generates circular RNAs, it also produces abundant aberrantly spliced linear by-products (Ho-Xuan et al, 2020), including *trans*-spliced RNAs arising from internal promoters and splicing between distinct molecules (Chu et al, 2021).

Despite these observations, routine plasmid-based experiments rarely examine the full diversity of RNA species generated by transgenes. Even when aberrant splicing is observed, it may be dismissed as experimental noise or left unreported, meaning that the published examples may only represent the tip of the iceberg. In the following section, we examine the common sources of unintended splice sites in transgene constructs.

### Translator's betrayal: codon optimization changes the script

While codon sequences are nearly universal across organisms, codon usage preferences differ significantly (Hanson and Coller, 2018). To boost transgene expression, researchers often apply codon optimization, introducing silent mutations to shift the mRNA sequence towards one optimized for the host cell's tRNA abundance (Fig. 3A,B). This approach assumes that the underlying RNA sequence is largely inconsequential and can be modified without affecting the protein product. However, synonymous mutations in coding sequences are associated with dozens of human diseases through effects on RNA processing, folding, and stability (Sauna and Kimchi-Sarfaty, 2011), underscoring that coding sequences also carry important regulatory information. Codon optimization can inadvertently introduce or alter RNA regulatory motifs, including splice sites, promoters, and polyadenylation signals (Bartys et al, 2019; Hunt et al, 2014) (Fig. 3B). In some cases, codon optimization can even generate unintended ORFs, as seen in codon-optimized versions of the papillomavirus protein E7, which produced a novel synthetic peptide not present in the wild-type virus (Lorenz et al, 2015).

To illustrate how codon optimization can affect splicing, we used the computational splice site prediction algorithm MaxEntScan (Yeo and Burge, 2004). The MaxEnt score for a sequence reflects the log-odds ratio of finding that sequence in authentic versus decoy human splice sites. Most functional splice sites have MaxEnt scores $\geq 0$, with higher MaxEnt scores indicating stronger splicing potential, although the score does not directly predict in vivo usage (Fig. 3C). As an example, the amino acid sequence Gln-Val-Ser can be encoded using suboptimal codons that do not contain a 5' splice site (5' MaxEnt = -13.92; Fig. 3C). However, codon optimization of this motif can create a near-cognate donor site (5' MaxEnt = 9.6; Fig. 3C), potentially enabling splicing if a 3' splice site is present downstream. Likewise, codon optimization can generate strong acceptor sites, such as in the sequence Pro-Pro-Pro-Gln (median 3' MaxEnt score = 6.7, depending on surrounding sequence; Fig. 3C). These motifs are not rare: Gln-Val-Ser and Pro-Pro-Pro-Gln appear in ~11% and 3% of proteins in the MANE Select human proteome (Morales et al, 2022), respectively. Many other common motifs similarly acquire splicing potential upon optimization (e.g., Ser-Val-Ser or Leu-Leu-Gly-Ser, and other examples, Fig. 3C).

In addition to creating new signals, codon changes can disrupt evolved features such as RNA secondary structures or splicing silencers that suppress aberrant processing of the native gene. For instance, codon optimization of a SARS-CoV-2 gene resulted in new splice sites and corresponding aberrantly spliced transcripts (Tomberg et al, 2021). While codon optimization remains a powerful tool for improving transgene expression, its potential to inadvertently introduce splicing signals and disrupt critical RNA regulatory elements underscores the need for careful design and validation of synthetic gene sequences.

### Unscripted cameos: vector elements as sources of unintended splice sites

Transgenes are embedded in non-native, plasmid-derived sequence contexts that can inadvertently introduce motifs recognized by the host's splicing machinery. Several commonly used vectors include introns in the 5' UTR to enhance nuclear export of the transcript (Hitoshi et al, 1991). Vectors like pCAG and pAAV use a synthetic CAG promoter, which combines a CMV promoter with a chimeric intron constructed from the 5' splice site of chicken β-actin and the 3' splice site of rabbit β-globin (Hitoshi et al, 1991) (Fig. 3A,C). While this design improves transcription and translation efficiency, it can also lead to unintended splicing events involving the transgene (Fallot et al, 2009; Peterman et al, 2025). For example, phenomena initially attributed to frameshifting in the *CCR5* coding sequence or an IRES in the *XIAP* 5' UTR were both later shown to result from cryptic splicing occurring between the chimeric intron and the transgene (Khan et al, 2022; Lemp et al, 2012).

Bacterial sequences within plasmid backbones can also harbor cryptic RNA processing signals, as they have not been evolutionarily selected for compatibility with eukaryotic gene expression (Lemp et al, 2012) (Fig. 3A,D). As a notable example, the ampicillin resistance cassette (AmpR), commonly used for plasmid selection in bacteria, contains both 5' and 3' splice sites that are recognized in mammalian cells (Lemp et al, 2012). If AmpR-derived sequences are transcribed through cryptic promoters, they can be spliced to the transgene (Lemp et al, 2012). In a recent study, we showed that transcription initiation from the ColE1 origin of replication

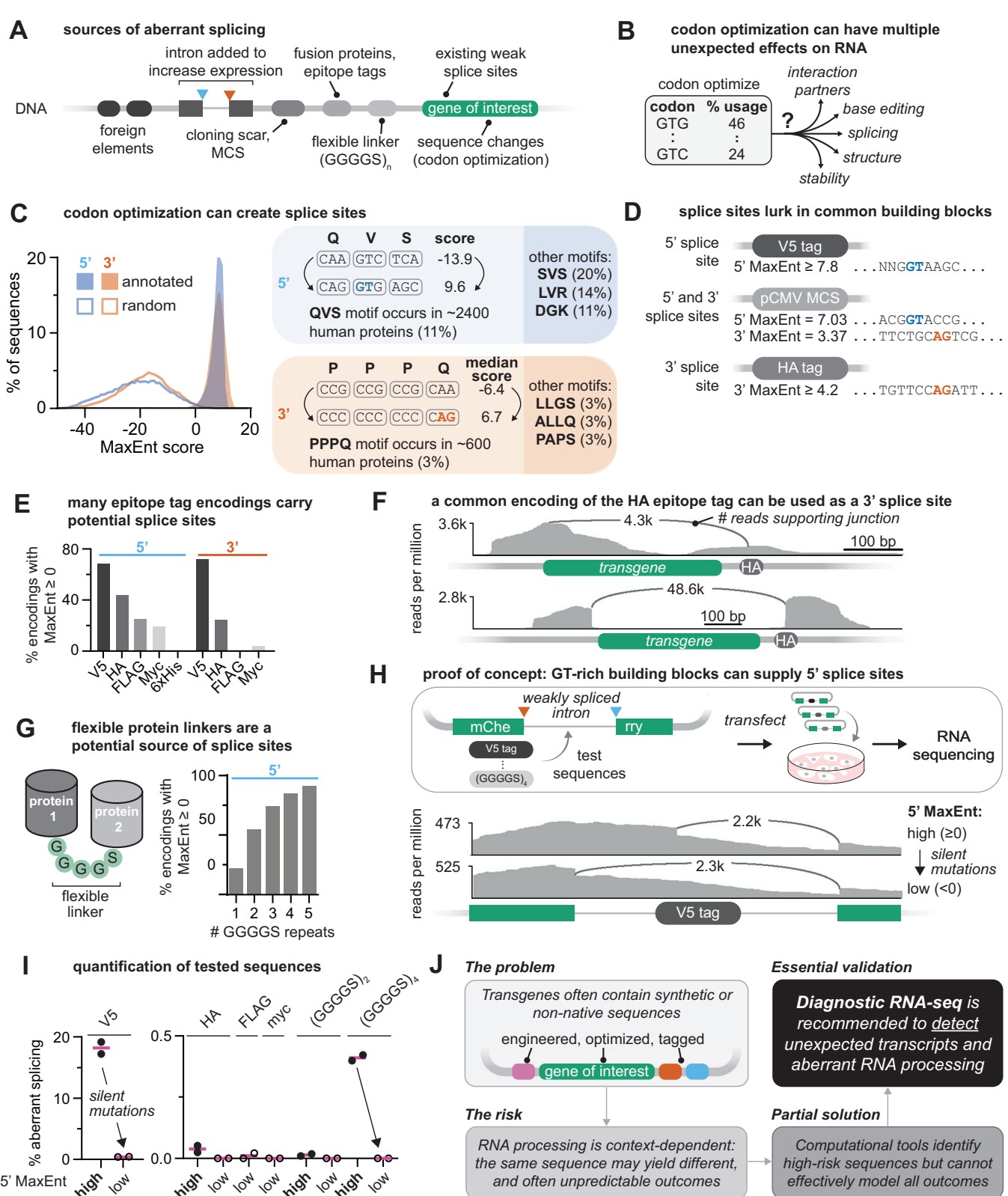

**A** sources of aberrant splicing

**B** codon optimization can have multiple unexpected effects on RNA

**C** codon optimization can create splice sites

**D** splice sites lurk in common building blocks

**E** many epitope tag encodings carry potential splice sites

**F** a common encoding of the HA epitope tag can be used as a 3' splice site

**G** flexible protein linkers are a potential source of splice sites

**H** proof of concept: GT-rich building blocks can supply 5' splice sites

**I** quantification of tested sequences

**J** The problem

*Transgenes often contain synthetic or non-native sequences*

engineered, optimized, tagged → gene of interest

*The risk*

*RNA processing is context-dependent: the same sequence may yield different, and often unpredictable outcomes*

*Partial solution*

*Computational tools identify high-risk sequences but cannot effectively model all outcomes*

*Essential validation*

**Diagnostic RNA-seq** *is recommended to* underline{detect} *unexpected transcripts and aberrant RNA processing*

resulted in transcription across the AmpR locus and into the transgene (Anderson et al, 2024). This ColE1 initiated pre-mRNA contained 5′ splice sites within AmpR that

were spliced to 3′ splice sites in the transgene, producing unexpected mRNAs that encoded spurious, out-of-frame proteins (Anderson et al, 2024) (see model in

Fig. 2B). Even though some vectors incorporate sophisticated elements like insulators and polyadenylation elements intended to minimize spurious promoter activity (Carey

**Figure 3. Drivers of aberrant splicing in transgenes.**

(A) Potential sources of splicing elements in plasmids, such as the origin of replication, multiple cloning site (MCS), built-in introns to increase expression, non-native sequence introduced during cloning, and cryptic splice sites in the transgene itself. (B) Codon optimization introduces synonymous mutations to match host codon usage, but can inadvertently disrupt evolved constraints on RNA structure, processing signals, and RNA-protein interactions, potentially creating unexpected vulnerabilities. (C) Codon optimization can create near-consensus splice sites by chance. *Left:* MaxEnt scores for all annotated human 5′ and 3′ splice sites (solid fills) and for 100,000 random sequences (lines). By definition, most human 5′ and 3′ sites (~97%) score $\geq 0$, but only a small fraction of random sequences do (~2% score $\geq 0$). *Right:* Examples of how codon optimization can introduce a novel splice site through synonymous changes into a sequence that lacked 5′ or 3′ splicing signals. (D) Synthetic building blocks can encode splice sites, depending on the codons used. Some V5 tag encodings produce strong 5′ splice sites (5′ MaxEnt score $\geq 7.8$), and HA tag encodings can contain moderate 3′ splice sites (3′ MaxEnt score $\geq 4.2$). Likewise, the pCMV multiple cloning site (sequence obtained from Addgene #23007) contains sites with 5′ and 3′ MaxEnt scores $\geq 0$. (E) Splicing potential of common epitope tags. For each tag, $10^6$ random codon encodings were generated, and the highest 5′ or 3′ MaxEnt score per sequence was plotted. The 6xHis tag is omitted from 3′ site analysis as it is short and lacks the essential 'AG' dinucleotide required to define a 3′ splice site. (F) Sashimi plot showing RNA-sequencing coverage and splicing of HA-tagged transgenes. Splice junction-spanning reads are indicated as arcs; read counts are shown above each arc. (G) Flexible linkers composed of Gly-Ser repeats can harbor splice sites. For each linker length, $10^6$ codon-randomized sequences were generated. Linkers were scored as "splicing-prone" if any 5′ site had MaxEnt $\geq 0$. These randomly generated repeats often contain potential splice sites due to the prevalence of "GT" dinucleotides in their codons. (H) *Top:* Design of a splicing reporter expressing mCherry split by a weakly spliced intron. Test sequences (e.g., a V5 epitope tag) were inserted into the intron to assess their potential to drive splicing in the presence of nearby weak splice sites. *Bottom:* Sashimi plot for the indicated V5 test sequences, as in (F). (I) Quantification of reads reflecting aberrant splicing for test sequences in the split-mCherry reporter, as a proportion of total reads across the splice junction and within the intron. Test sequences were binned into high ($\geq 0$) or low ($<0$) MaxEnt score based on the highest 5′ MaxEnt score present in the sequence. (J) Summary schematic illustrating the risks of using heavily modified plasmid models and potential strategies to mitigate them. While computational tools and careful sequence design can help reduce unintended RNA processing, a diagnostic RNA-seq experiment is strongly recommended to validate the transcripts generated by model systems.

et al, 2009) (e.g., pGL4, pTwist), these constructs are still prone to generating aberrantly spliced transcripts (Lemp et al, 2012).

Similarly, regions such as multiple cloning sites (MCS), designed for flexible "cut-and-paste" cloning, were not developed with mammalian RNA processing in mind and may inadvertently contain splicing motifs (Fig. 3A,D). The widely used pCMV plasmid contains a 78-nucleotide MCS with several restriction sites, including KpnI. By chance, the KpnI recognition sequence (GGTACC) is predicted to produce a putative 5′ splice site in most sequence contexts (98% of NNGGTACCN combinations yield a MaxEnt score $\geq 0$, where N is any nucleotide, Fig. 3D). The pCMV MCS also harbors a potential 3′ splice site (3′ MaxEnt = 3.37, Fig. 3D). In one report, the NheI restriction sequence (GCTAGC) was able to act as a 3′ splice site by providing the critical "AG" dinucleotide to a pyrimidine-rich region in the transgene (Murauer et al, 2013). If these elements are retained during cloning, they will be included in the transcript and could promote unintended splicing. Importantly, these spurious splice sites may not be used uniformly across all transcripts. Their stochastic usage can lead to the production of multiple RNA isoforms, some of which may interfere with intended outcomes. This transcript diversity can make spurious events difficult to detect, obscure experimental readouts, and complicate troubleshooting, especially when unintended variants co-exist with the desired transcript.

## When props malfunction: protein tags and linkers

Splice sites can also be found in commonly used protein tags, such as fluorescent proteins, eukaryotic antibiotic resistance cassettes, and epitope tags (Fig. 3A). Aberrant splicing and resulting chimeric transcripts involving the transgene have been reported in constructs containing ampicillin (Lemp et al, 2012; Muerdter et al, 2018; Anderson et al, 2024), neomycin (Cheng et al, 2022; Roshon et al, 2003), and puromycin (Cheng et al, 2022) resistance cassettes as well as in fusion proteins, such as Renilla luciferase (Ansseau et al, 2015), GFP (Majerciak and Zheng, 2016; Gutierrez-Triana et al, 2016), and mKate2 (Peterman et al, 2025). Even short tags can harbor splice sites. For instance, one encoding of the V5 epitope tag, derived from simian virus 5, contains a strong 5′ splice site (5′ MaxEnt $\geq 7.8$, see Supplemental Information for sequence, Fig. 3D). When placed at the N-terminus of proteins, both the V5 sequence and portions of the transgene have been found to be spliced out (Ansseau et al, 2015; Cheng et al, 2022).

Importantly, whether a tag introduces splice sites depends on its nucleotide sequence. Alternative codon choices for the same amino acid sequence can eliminate these motifs. In our analysis of common epitope tags, we found that many potential V5 and HA codon variants contain putative 5′ and 3′ splice sites, with over 75% of V5 encodings containing both (Fig. 3E). Critically, these splice-prone sequences are

widely used: a V5 encoding with strong splice sites is included in the popular molecular biology software SnapGene. Similarly, an HA encoding with a 3′ splice site (3′ MaxEnt $\geq 4.2$, see Supplemental Information for sequence, Fig. 3F) appears in at least 44 published articles and is present in several "empty" plasmid backbones available from the Addgene repository (e.g., plasmid #128034 (pcDNA 3.1-HA), which has been cited 23 times at the time of this writing). By contrast, epitope tags such as FLAG, Myc, and 6xHis are less likely to introduce splice sites because their potential encodings lack critical 'GT' and 'AG' dinucleotides (Fig. 3E). Notably, splicing-mediated loss of these tags is difficult to detect, as the tag is removed from the mature protein, and aberrantly spliced protein variants may go unnoticed (Ansseau et al, 2015; Cheng et al, 2022).

Protein linkers, which are often added to fusion proteins to maintain structural flexibility and prevent steric clashes between folded domains, can also introduce splice sites. A common example is the glycine-serine linker (Gly-Gly-Gly-Gly-Ser), often repeated several times to space out folded domains (Chen et al, 2013) (Fig. 3A,G). These linkers are frequently encoded using suboptimal codons, in order to introduce sequence complexity, lower GC content, and facilitate cloning. However, Gly and Ser codons often contain the "GT" dinucleotide, a key feature of 5′ splice sites (e.g., Gly-Gly-Ser motif encoded by DNA sequence GGA GGT AGT N where N is any base, has 5′ MaxEnt score $\geq 4.45$; Fig. 3G). When

multiple Gly-Gly-Gly-Gly-Ser motifs are incorporated, the likelihood of generating spurious splice sites increases significantly. Notably, one study reported that removing 'GT' motifs in a (Gly-Gly-Gly-Gly-Ser)$_3$ linker dramatically increased RNA abundance and protein expression (Trinh et al, 2004). We note that currently there are no published reports of linker-associated splicing artifacts, but this absence may merely reflect a lack of focused investigation.

The potential for these elements to drive aberrant splicing can be illustrated in a reporter system where various test elements were inserted into a weakly spliced intron (see Supplemental Information for experimental details, Fig. 3H). Here, the common V5 epitope tag with high 5' MaxEnt scores (described above) underwent splicing in ~18% of reads (Fig. 3H,I). A long Gly-Ser linker also supported low levels of splicing (~0.4% of splice junctions involving the linker, Fig. 3I). Although modest, these results underscore that GT-rich linkers can serve as 5' splice sites in permissive sequence environments when a compatible 3' splice site is present. By contrast, common encodings of the FLAG and Myc tags, which lack predicted splice sites (MaxEnt scores ≥0), did not show detectable splicing (Fig. 3I). Importantly, silent mutations that eliminated high-scoring splice sites (MaxEnt ≥0) in both V5 tags and flexible linkers completely abolished aberrant splicing, demonstrating that simple computational sequence optimization can minimize splicing artifacts (Fig. 3H,I). The context-dependent nature of splicing was further illustrated by an HA encoding that, despite containing a predicted 3' splice site, showed no splicing activity in this reporter system (Fig. 3I). The prevalence of splice sites in common plasmid elements, coupled with their context-dependent, unpredictable, and often undetectable effects, presents a sobering reality: many transgene expression experiments may be compromised by artifacts hiding in plain sight. These vulnerabilities in our most basic molecular biology tools call for a fundamental shift in how we design, validate, and interpret transgene expression experiments (Fig. 3J).

## Directing a cleaner performance: best practices to minimize artifacts in transgene expression systems

The challenges outlined above, from cryptic promoters to aberrant splicing, demonstrate that plasmid-based transgene expression is far more complex than typically assumed. In this section, we offer practical recommendations for designing and validating experiments involving transgenes in order to minimize these pervasive artifacts and improve experimental reliability.

1. **Exercise caution when altering protein sequences:** Codon optimization is routinely marketed as a simple way to boost expression, but this seemingly innocuous practice can fundamentally alter experimental outcomes. Each nucleotide change risks introducing cryptic splice sites, internal promoters, polyadenylation signals, or other regulatory elements. Similar risks apply to epitope tags and flexible linkers. For native genes expressed in their cognate systems (e.g., mammalian genes in mammalian cells), carefully weigh whether the potential gains of sequence alterations outweigh the risk of creating unintended transcripts. When adding tags or linkers, choose previously validated sequences or those engineered to avoid problematic motifs (see below and Dataset EV1). Non-native and synthetic sequences require special attention, as they may harbor unintended RNA processing signals.

2. **Use computational tools to evaluate sequence designs:** Computational tools can identify many RNA processing signals prior to synthesis. For example, the Maximum Entropy model (MaxEntScan) (Yeo and Burge, 2004) provides a simple and easily interpretable scoring system for putative splice sites. High-scoring sequences (≥0) can be tweaked to remove the crucial 'GT' and 'AG' dinucleotides required for splicing. Similar tools can identify cryptic promoter elements and polyadenylation signals. We provide DNA sequences for common synthetic building blocks (Adames et al, 2015) that are computationally optimized to minimize splicing potential (see Dataset EV1), although their effectiveness in avoiding undesired RNA processing should be experimentally verified in the relevant cellular context. While current tools cannot predict all RNA processing outcomes, they provide valuable first-pass screening. As the tools available in this space continue to advance (Jaganathan et al, 2019), they may enable more sophisticated sequence engineering to minimize unintended processing.

3. **Publish complete plasmid sequences:** For transgene-based studies, the plasmid represents the source code of the experiment, yet full plasmid sequences are rarely made available. Without the full sequence, results cannot be reliably interpreted, reproduced, or compared across studies. This information is invaluable for interpreting and reproducing results, particularly when unexpected molecular or phenotypic effects are observed. Mandatory deposition of complete plasmid sequences (and whenever possible, corresponding RNA-seq data for key models) would greatly enhance transparency, reproducibility, and biological rigor. It would enable reanalysis of existing data as our understanding of RNA processing and transgene behavior evolves, and would allow systematic identification of problematic constructs across laboratories and studies (Loughran et al, 2025; Peccoud et al, 2011; Thuronyi et al, 2023). This, in turn, would save considerable community resources by flagging reagents prone to artifacts and preventing the propagation of misleading results.

4. **Confirm causality with redundancy and rescue across orthogonal systems:** While plasmids remain indispensable molecular biology tools, they are imperfect models of endogenous gene expression. When direct analysis of the endogenous gene is not feasible, consider validating key findings across multiple expression systems (e.g., transient expression vectors, viral or transposon delivery, knock-in cell lines, or transfection of in vitro transcribed RNA). Although direct RNA transfections can mitigate issues with cryptic promoters and splicing, in vitro transcription is not artifact-free: it may produce truncated isoforms or dsRNA (Lenk et al, 2024), and the resulting RNA lacks co-transcriptionally deposited RNA-binding proteins, potentially distorting RNA stability, localization, and translation. Because there are no flawless expression systems, rescue experiments are essential. The phenotype should be tested with mechanism-appropriate controls, such as comparing the wild-type protein to a catalytic-dead or domain-mutant version. Other key tests, as relevant, include using a synonymous recode of the transgene to disrupt any potential RNA-level artifacts (e.g., cryptic splice sites), an ORF-frameshift mutant to probe for RNA-only effects, or point mutants designed to

disrupt potential internal translation initiation sites (Loughran et al, 2025). Although each approach has inherent limitations (Andreev et al, 2016), consistent results across different systems supported by robust controls can strengthen confidence in biological relevance while minimizing the risk of system-specific artifacts.

5. **Use RNA sequencing to assess transcript fidelity:** Even carefully designed and computationally screened constructs can generate unintended RNA products. Targeted assays, such as Western blots, fluorescent protein markers, antibiotic resistance markers, or diagnostic PCR amplification of the gene of interest, are not suited for this task. They are blind to mis-processing events that remove the very diagnostic features (like an epitope tag or primer site) they are designed to measure, nor can they reliably assess low-frequency variants. As a result, potentially biologically meaningful aberrations, including rare splice variants or cryptic transcription products, may go undetected. We strongly recommend that unbiased RNA sequencing be used to assess the full repertoire of RNAs produced from key transgenes, particularly in light of unexpected or extraordinary results. The sequencing reads should be mapped to the entire plasmid sequence, including the backbone, to capture products of cryptic promoters, aberrant splicing, premature polyadenylation, or other RNA processing artifacts. Because most plasmids use strong promoters, informative coverage can often be achieved at modest sequencing depths. Sample multiplexing could further reduce costs to levels comparable to conventional Sanger or Nanopore-based plasmid validation methods ($5–15/sample). Adopting these practices is critical to ensuring that published findings reflect true biological insights rather than unintended consequences of transgene design.

The unwritten standards for reporting on plasmids in research papers have remained largely unchanged since their inception. While authors are generally expected to describe the cloning steps involved in plasmid construction, they are not required to provide full plasmid sequences or demonstrate that upon incorporation into the host, the transgene DNA (only) produces the intended RNA sequence. We argue that it is imperative to evaluate all products arising from transgene expression, not just those expected to result from the expressed gene. The expected controls and reporting standards for transgenic models should be updated in light of our current understanding of RNA biology, as well as the relatively low cost and accessibility of RNA sequencing.

Inadequate reagent validation is a major contributor to the reproducibility crisis in the biological sciences, leading to issues such as cell line misidentification (Lucey et al, 2009), mycoplasma contamination (Olarerin-George and Hogenesch, 2015), unreliable antibodies (Bradbury and Plückthun, 2015), and errors in plasmid assembly (Bai et al, 2025). The US National Institutes of Health and many journals now include cell line authentication as a prerequisite for grant applications and publication. We argue that journals and funding agencies should mandate that all published plasmids and transgenes include their complete sequence in supplemental data, along with RNA sequencing data for key models, to ensure rigorous and reproducible results. With the increasing availability of RNA sequencing services, this verification poses a minimal burden to researchers. Many companies now offer library preparation and sequencing as a service, and the cost per sample continues to decline, averaging around $125 per sample at the time of writing. This is a fraction of the cost of a standard commercial antibody, and can save countless hours of personnel time and valuable resources by preventing the pursuit of erroneous leads and the propagation of inaccurate findings.

As scientists, we are naturally drawn to intriguing and unexpected results. Once such data emerge, there is a strong drive to establish their biological relevance or to connect them with disease mechanisms. In today's collaborative and fast-paced research landscape, these findings can rapidly gain momentum, with a given transgenic system rapidly becoming an established "model". Before long, these data form the foundation for intellectual property claims, funding proposals, biotechnology ventures, and even clinical trials. For decades, it has been known that virtually every RNA in human cells undergoes extensive processing before it encounters the translation machinery. Yet, it is assumed, without scrutiny, that exogenous sequences introduced via plasmids faithfully produce cognate mature RNAs matching the input sequence. This oversimplification has misled the scientific community on multiple occasions, allowing artifacts to masquerade as true biological phenomena. While science is inherently self-correcting, it is imperative that we must learn from our past mistakes and implement systematic guardrails that account for the intrinsic complexities of RNA processing, ensuring that transgenic systems faithfully recapitulate biological reality.

## Peer review information

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

## Acknowledgements

We would like to thank the members of the Jain lab for their valuable feedback and discussions. This work was supported by grants from the National Institutes of Health (R35GM151111), the Chan Zuckerberg Initiative (DAF2022-250422), and the David and Lucile Packard Foundation.

## Author contributions

**Rachel Anderson**: Conceptualization; Resources; Data curation; Software; Formal analysis; Validation; Investigation; Methodology; Writing—original draft; Writing—review and editing. **Christalyn Ausler**: Resources; Data curation; Formal analysis. **Ankur Jain**: Conceptualization; Supervision; Funding acquisition; Writing—original draft; Project administration; Writing—review and editing.

## Disclosure and competing interests statement

The authors declare no competing interests.

