## [Peer Review File · The EMBO Journal]

When RNA goes off script: ensuring transcript fidelity in transgene expression

Rachel Anderson, Christalyn Ausler, and Ankur Jain

Corresponding author(s): Ankur Jain (ajain@wi.mit.edu)

Review Timeline:

Submission Date:	16th Jun 25
Editorial Decision:	29th Jul 25
Revision Received:	17th Nov 25
Accepted:	2nd Feb 26

Editor: Cornelius Schneider

Transaction Report:

Dear Dr. Jain,

Thank you for submitting your manuscript for consideration by the EMBO Journal. It has now been seen by three referees whose comments are enclosed. As you will see, all three referees express interest in your manuscript and are broadly in favor of publication. I find their suggestions helpful and productive and would like to invite you to submit a revised version of the manuscript, addressing the comments of all three reviewers. I think that the inclusion of experimental data is beneficial here but agree with referee #2 that it would be helpful if the representation would be easier to understand.

We generally allow three months as standard revision time. As a matter of policy, competing manuscripts published during this period will not negatively impact on our assessment of the conceptual advance presented by your study. However, we request that you contact the editor as soon as possible upon publication of any related work, to discuss how to proceed.

Thank you for the opportunity to consider your work for publication. I look forward to your revision.

Yours sincerely,

Cornelius Schneider, PhD
Editor
The EMBO Journal
c.schneider@embojournal.org

We realize that it is difficult to revise to a specific deadline. In the interest of protecting the conceptual advance provided by the work, we recommend a revision within 3 months (27th Oct 2025). Please discuss the revision progress ahead of this time with the editor if you require more time to complete the revisions. Use the link below to submit your revision:

Referee #1:

This is very well-written perspective that provides a very careful and thoughtful consideration of many of the complexities that must be considered for transgene expression.

In general, I am very supportive of this piece.

I have several minor comments / suggestions.

Page 5 line 1

It is more accurate to say 'apparent ribosomal frameshifting'

Page 7 last sentence of first paragraph

'Notably, many human genes naturally harbor downstream AUG codons that support regulated expression of alternative isoforms²³, which may interfere with the function of the full-length protein²⁴.'

I suggest including this reference (PMID 32669370) here which shows that the first AUG 3' of the annotated start codon is more likely to be in the same reading frame.

Page 7

'In fusion constructs, these internally initiated transcripts can give rise to reporter-positive fragments that confound interpretation by decoupling the reporter signal from the full-length transgene.'

I think this sentence is misleading. The internal promoter activity discovered in firefly luciferase produces transcripts that have no luciferase activity as they do not encode the N-terminal region required for reporter activity. I agree that for some types of assay (perhaps immunoblotting or immunofluorescence) this internal promoter activity could lead to false positives but not when the end point is an assay measuring the enzymatic activity of firefly luciferase.

Page 11

'In eukaryotes, nascent transcripts undergo extensive RNA processing, including 5' capping, splicing, 3' end cleavage, polyadenylation, base modification, and nuclear export. These steps are guided by short, degenerate sequence motifs that are shaped by contextual rules that remain incompletely understood.'

5' capping is not guided by short, degenerate sequence motifs that are shaped by contextual rules. I suggest changing the first sentence to something like 'In eukaryotes, capped nascent transcripts undergo extensive RNA processing, including splicing, 3' end cleavage, polyadenylation, base modification, and nuclear export.'

Page 12

The ATP7B artefact described in reference 12 was not a splicing artefact.

Page 24 line 3

Shouldn't only mammalian expression plasmids be listed here?

Referee #2:

The manuscript "When RNA goes off-script: misadventures in transgene expression" provides a useful and important reminder that recombinant DNA in the cellular context may have a broad range of unintended and often not examined effects. Authors an organized overview of possible artifacts, which includes published results, where expression of recombinant DNA went wrong and yielded various artifacts. A part of the work includes original experimental data to document splicing artifacts arising from GT-rich protein linkers. Authors also provide a list of recommendations to address the possible issues.

Overall, I find the article an important and timely reminder for anyone working with recombinant DNA to be careful and alert as undesirable artifacts may easily appear in experiments and mislead their interpretation. While the majority of the possible issues may be obvious and makes common sense, I suspect that most experiments with plasmid reporters take place with little if any attention to possible artifacts unless unexpected results appear, which would prompt further analysis.

I have several comments to consider when revising the manuscript:

Major point:

1) In my view, inclusion of experimental data does not fit the perspective format and these data not necessary to convey the message of the perspective. Experimental data included in the manuscript disrupt the reading as one has to go to the methods to try to understand what the Figure 3I means and how was it constructed while the only point is that, in a specific context, GU-rich regions in transcripts are potential cryptic splice sites. In a perspective article, this would be sufficient to comment or show a single sequencing data snapshot like in the bottom of the panel 3H and extend the figure legend accordingly.

Minor points

1) The end of the abstract sounds odd and is confusing, the last sentence especially. Please, consider revising.

2) page 3 - Do authors really mean "animal models" or could this be "organisms"?

3) It could be mentioned that cryptic promoters are also present in the genomic DNA and sequence changes. Consequently, a transgene insertion or sequence deletion may activate them as these manipulations might disrupt the local chromatin, enhancer, and/or transcriptional landscapes, which otherwise would silent such cryptic promoters.

4) Concatemerization during insertion of transgenes is another issue, which could be mentioned as a potential source of artifacts. Tandem arrays of transgenes may contain inverted repeats and give rise to dsRNA, which in turn may have various effects.

5) Polyclonal populations of stably integrated transgenes may provide a false sense of safe conduct while their longer culture is actually vulnerable to clonal artifacts. Clonal effects in this situation may possibly represent even bigger problem as they would come from a faster growth phenotype and may be more difficult to monitor/control in contrast to clonally-selected lineages where redundancy of phenotypes of independent lineages allows to sort out insertion site effects. Also, in clonal lineages, one may be able to identify insertion sites with inverse or splinkerette PCR and then monitor transcriptome changes in that locus.

6) Among the practical considerations should be careful experimental design - perhaps this should be even added as a specific point given importance of it. Authors touch upon these in the point 4 (validating finding using different strategies/systems) but I think the point could be discussed more, including concepts of redundancy and rescue strategies- it may be impossible to control all possible artifacts of transgenic expression but carefully designed rescue experiments may allow to demonstrate that effects/phenotypes are coming from intended experimental manipulation and not from some sort of an artifact.

7) My personal view which authors may or may not consider: I understand the request that primary authors should completely sequence their plasmids. At the same time, I am concerned with ever increasing demands on primary authors who generate primary reagents and make them publicly available to everyone else in good faith. Plasmid sequencing and verification may be done as well by reagent repositories such as Addgene (who actually do it) and/or downstream users. Primary authors should not be forced to analyze plasmid-borne transcriptomes beyond their own experimental design and peer review process.

Referee #3:

Plasmid based expression of transgenes is a foundational technology used in molecular biology, cell biology, genetics, genomics, and biotechnology. Included with this widespread use is the assumption that transfected cells express only the protein designed by the researcher. However, numerous studies have shown that transgenes often produce unexpected products. Despite being well-documented, the production of these unexpected products is almost entirely ignored by many researchers. This is a major problem for biomedical research, as much time, money, and effort is spent following false leads caused by transgene "misadventures". This manuscript is important in that it would highlight the issue for a wider audience of molecular biologists. It is, for the most part, a review of the numerous studies that have found such artifacts, with the addition of an example of splicing artifacts that can occur in a reporter library in Figure 3. Overall, I believe this would make a very nice review article that would be well-appreciated by the EMBO community of readers. I don't think there's enough new results if this was to be considered as a research article. I have a few comments that I believe should be addressed.

1. The authors should add a paragraph or two discussing circRNA plasmids that use inverted Alu repeats. Developed by Jeremy Wilusz's lab (PMID 25281217, 2014), this is designed to express circular RNA through "backsplicing" of a downstream 5' splice site to an upstream 3' splice site. This plasmid was used to claim that hundreds of mammalian genes have IRESes in 2021 (PMID 34437836) and to argue for IRES activity from the promoter regions of Hoxa9 and other genes this Spring (PMID: 40082722). However, the vast majority of RNAs expressed from this plasmid are linear and result from trans-splicing of 5' and 3' splice sites from separate mRNA molecules generated from internal promoters (PMIDs 33940551, 32955563, 33762403). Indeed, Dr. Wilusz wrote a 2021 review describing this artifact (PMID 33629517). This is an important additional example of another artifact type (trans-splicing) and highlights the fact that most researchers are not aware of these issues, even years after they are published.

2. An excellent example of splicing artifacts has been posted as a preprint in biorxiv (PMID: 39149310). In this case, cryptic splicing was found to explain the results of multiple 3' UTR reporter libraries, including one authored by Sergej Djuranovic in 2018. Dr. Djuranovic has admirably posted a followup analysis showing these effects. This should be included in the manuscript.

3. The authors advocate requiring full-length plasmid sequence publication and RNA-seq validation of plasmids. I think this is a wonderful idea. However, wouldn't it be better for researchers to avoid these issues by using RNA transfections instead of plasmid transfections? Instead of finding out after the fact that an apparently "exciting" result was an artifact of cryptic promoters or splicing, losing several months of work along the way, direct RNA transfections would allow researchers to get more accurate results the first time around. It would also avoid temptation to malpractice.

4. It would be fantastic for EMBO to lead the way in this by requiring authors to publish full plasmid sequences, and requiring (or at least strongly encouraging) RNA-seq validation of plasmids or RNA transfection validation, especially when plasmid transfection experiments are used to make extraordinary claims. They could also ask reviewers to consider whether plasmid transgenes could be making unintended products (linked to this and other reviews on the subject).

Note: For ease of comparison, revised portions of the manuscript text are *indicated in italics*.

Referee comments

Referee #1: This is very well-written perspective that provides a very careful and thoughtful consideration of many of the complexities that must be considered for transgene expression.

In general, I am very supportive of this piece.
I have several minor comments / suggestions.

Author response: We thank the reviewer for making time to review our work and for providing valuable feedback.

Page 5 line 1 - It is more accurate to say 'apparent ribosomal frameshifting'

Author response: This is a good point. We have changed the wording to “...*apparent ribosomal frameshifting* in CCR5¹¹ ...”

Page 7 last sentence of first paragraph

'Notably, many human genes naturally harbor downstream AUG codons that support regulated expression of alternative isoforms²³, which may interfere with the function of the full-length protein²⁴.'

I suggest including this reference (PMID 32669370) here which shows that the first AUG 3' of the annotated start codon is more likely to be in the same reading frame.

Author response: Thank you for this suggestion which clarifies our concerns. We have added the reference and expanded the text: “Notably, many human genes naturally harbor additional *in-frame* AUG codons downstream from the annotated translation start site; *initiation at these sites allows regulated expression of alternative isoforms (e.g., by removing localization signals), that can compete with or modulate the function of the full-length protein*^{24–26}.”

Page 7

'In fusion constructs, these internally initiated transcripts can give rise to reporter-positive fragments that confound interpretation by decoupling the reporter signal from the full-length transgene.'

I think this sentence is misleading. The internal promoter activity discovered in firefly luciferase produces transcripts that have no luciferase activity as they do not encode the N-terminal region required for reporter activity. I agree that for some types of assay (perhaps immunoblotting or immunofluorescence) this internal promoter activity could lead to false positives but not when the end point is an assay measuring the enzymatic activity of firefly luciferase.

Author response: We agree that the wording was misleading and did not make it clear we were broadly referring to internal promoters. We have updated the text for clarity: “For instance, the widely used firefly luciferase, a staple in gene expression assays, harbors a cryptic internal promoter capable of initiating transcription in both yeast and mammalian cells²¹. *More broadly, in fusion constructs, internal initiation events can give rise to unexpected protein products in alternative reading frames or even generate*

reporter-positive fragments that decouple the reporter signal from the full-length transgene, thus confounding interpretation.”

Page 11

'In eukaryotes, nascent transcripts undergo extensive RNA processing, including 5' capping, splicing, 3' end cleavage, polyadenylation, base modification, and nuclear export. These steps are guided by short, degenerate sequence motifs that are shaped by contextual rules that remain incompletely understood.'

5' capping is not guided by short, degenerate sequence motifs that are shaped by contextual rules. I suggest changing the first sentence to something like 'In eukaryotes, capped nascent transcripts undergo extensive RNA processing, including splicing, 3' end cleavage, polyadenylation, base modification, and nuclear export.'

Author response: Thank you for pointing this out, we have updated the text: “In eukaryotes, nascent transcripts undergo extensive processing, including splicing, 3' end cleavage, polyadenylation, base modification, and nuclear export. *Many of these steps are guided by weakly conserved sequence motifs that are interpreted in a context-dependent manner.*”

Page 12

The ATP7B artefact described in reference 12 was not a splicing artefact.

Author response: Thank you, we removed the improper citation.

Page 24 line 3

Shouldn't only mammalian expression plasmids be listed here?

Author response: We understand that this might seem out of place. However, we want to highlight bacterial expression and cloning plasmids as a potential risk because they are sometimes used to generate new mammalian expression systems (e.g. CMV promoters and transgenes were added to pUC plasmids for Addgene #42230 and #227642) which would then have spurious promoter activity.

We have revised the panel legend to better separate the plasmids by use case, and to more clearly emphasize this risk factor: “Cryptic promoter activity can also arise from plasmid elements such as the ColE1-family origins of replication which are ubiquitous in modern plasmids (*e.g. in existing mammalian expression systems like pCMV, pcDNA, pDest26, pHR, and their derivatives, as well as in bacterial expression and cloning plasmids such as pBluescript, pUC, and pBR322 when re-purposed for use in mammalian systems*)⁸⁹.”

Referee #2:

The manuscript "When RNA goes off-script: misadventures in transgene expression" provides a useful and important reminder that recombinant DNA in the cellular context may have a broad range of unintended and often not examined effects. Authors an organized overview of possible artifacts, which includes published results, where expression of recombinant DNA went wrong and yielded various artifacts. A part of the work includes original experimental data to document splicing artifacts arising from GT-rich protein linkers. Authors also provide a list of recommendations to address the possible issues.

Overall, I find the article an important and timely reminder for anyone working with recombinant DNA to be careful and alert as undesirable artifacts may easily appear in experiments and mislead their interpretation. While the majority of the possible issues may be obvious and makes common sense, I suspect that most experiments with plasmid reporters take place with little if any attention to possible artifacts unless unexpected results appear, which would prompt further analysis.

Author response: We thank the reviewer for making time to review our paper, highlighting the key points from our work, and for providing useful feedback.

I have several comments to consider when revising the manuscript:

Major point:

1) In my view, inclusion of experimental data does not fit the perspective format and these data not necessary to convey the message of the perspective. Experimental data included in the manuscript disrupt the reading as one has to go to the methods to try to understand what the Figure 3I means and how was it constructed while the only point is that, in a specific context, GU-rich regions in transcripts are potential cryptic splice sites. In a perspective article, this would be sufficient to comment or show a single sequencing data snapshot like in the bottom of the panel 3H and extend the figure legend accordingly.

Author response: Thank you for this feedback. We realize that it is unusual to include new data in a perspective article and thus have simplified Fig. 3H-I. Specifically, we moved experimental details to a supplemental note and simplified the figure panels to exclude non-essential details:

“The potential for these elements to drive aberrant splicing can be illustrated in a reporter system where various test elements were inserted into a weakly spliced intron (see Supplementary Note 1 for experimental details, Fig. 3H).”

Minor points

1) The end of the abstract sounds odd and is confusing, the last sentence especially. Please, consider revising.

Author response: Thank you for this feedback. We agree that the phrasing was strained and have omitted the final two sentences:

“... Finally, we call for a revision of community standards for experiments using transgenes: deposit complete plasmid sequences and verify the resulting transcripts using RNA-seq.”

2) page 3 - Do authors really mean "animal models" or could this be "organisms"?

Author response: We have corrected the wording: “...across a wide range of cell types and *organisms*.”

3) It could be mentioned that cryptic promoters are also present in the genomic DNA and sequence changes. Consequently, a transgene insertion or sequence deletion may activate them as these manipulations might disrupt the local chromatin, enhancer, and/or transcriptional landscapes, which otherwise would silence such cryptic promoters.

Author response: Thank you for this suggestion. Although we do discuss cryptic promoters within transgenes, we had overlooked a discussion of chromatin effects. We have expanded this portion of the manuscript:

“The transgene may harbor promoter-like sequences that can initiate transcription downstream of the intended start site^{13,16,20–22} (Fig. 1A). Eukaryotic genomes have many cryptic transcription initiation sites that are silenced by chromatin-associated regulatory mechanisms²³. However, plasmid-expressed transgenes and their surrounding synthetic sequences may lack critical regulators, leading to re-activation of these initiation sites.”

4) Concatemerization during insertion of transgenes is another issue, which could be mentioned as a potential source of artifacts. Tandem arrays of transgenes may contain inverted repeats and give rise to dsRNA, which in turn may have various effects.

Author response: This is an interesting point we had not considered, and we have expanded the manuscript to include a discussion of this behavior:

“Stable integration can also be achieved by transfecting linearized DNA, such as during knock-in experiments, but this method introduces its own set of potential artifacts. Insertions can occur at multiple loci across the genome, or as tandem arrays within a single locus. Within these arrays, residual bacterial backbone or truncated fragments of the expression cassette may persist. Individual copies may be inverted, form head-to-tail concatemers, or recombine with other homologous genomic sequences^{57,58}. These integration details are rarely characterized, and each of these factors, from copy number to orientation, can result in transcripts that deviate substantially from the intended design.”

5) Polyclonal populations of stably integrated transgenes may provide a false sense of safe conduct while their longer culture is actually vulnerable to clonal artifacts. Clonal effects in this situation may possibly represent even bigger problem as they would come from a faster growth phenotype and may be more difficult to monitor/control in contrast to clonally-selected lineages where redundancy of phenotypes of independent lineages allows to sort out insertion site effects. Also, in clonal lineages, one may be able to identify insertion sites with inverse or splinkerette PCR and then monitor transcriptome changes in that locus.

Author response: This is a good point. We have added a discussion of this topic:

“Accordingly, studies using monoclonal lines should incorporate replicate clones and transcript-level validation to avoid misattributing effects to the intended transgene. Polyclonal pools are not entirely refractory to these issues. Transgenic insertions may impart a growth or survival advantage, and during prolonged culture, these rapidly-dividing cells can dominate the population⁵⁹. For instance, in lineage tracking experiments performed on K562 cells, a small proportion of clones (~10) made up half of the population after 90 doublings⁶⁰. Although techniques exist to monitor clonality (e.g., lineage tracking or mapping insertion sites by splinkerette PCR⁶¹), limiting the use of high-passage transgenic cell lines may help reduce the risk of clonal sweeps.”

6) Among the practical considerations should be careful experimental design - perhaps this should be even added as a specific point given importance of it. Authors touch upon these in the point 4 (validating finding using different strategies/systems) but I think the point could be discussed more, including concepts of redundancy and rescue strategies- it may be impossible to control all possible artifacts of transgenic expression but carefully designed rescue experiments may allow to demonstrate that effects/phenotypes are coming from intended experimental manipulation and not from some sort of an artifact.

Author response: This is a good suggestion, thank you. We have expanded point 4 with additional suggestions for controls:

“The phenotype should be tested with mechanism-appropriate controls, such as comparing the wild-type protein to a catalytic-dead or domain-mutant version. Other key tests, as relevant, include using a synonymous recode of the transgene to disrupt any potential RNA-level artifacts (like cryptic splice sites), an ORF-frameshift mutant to probe for RNA-only effects, or point mutants designed to disrupt potential internal translation initiation sites³⁶.”

7) My personal view which authors may or may not consider: I understand the request that primary authors should completely sequence their plasmids. At the same time, I am concerned with ever increasing demands on primary authors who generate primary reagents and make them publicly available to everyone else in good faith. Plasmid sequencing and verification may be done as well by reagent repositories such as Addgene (who actually do it) and/or downstream users. Primary authors should not be forced to analyze plasmid-borne transcriptomes beyond their own experimental design and peer review process.

Author response: We appreciate the reviewer’s thoughtful perspective and share the concern about placing undue burden on groups who generate and share primary reagents. Our intent is not to mandate comprehensive transcriptomic analyses for every construct, but to encourage proportionate transparency where it most affects interpretation.

We agree that repositories (e.g., Addgene) and downstream users play an important role in verification; however, deposition is not universal. As a low-burden baseline, we recommend that authors provide complete plasmid DNA sequences (insert and backbone) with the manuscript or via a public repository. In practice, these sequences are typically available during construct design and capture many of the issues we highlight.

RNA seq, however, imposes additional burden and may be unnecessary for routine experiments that are not central to the paper’s conclusions. Accordingly, we softened our guidance to recommend sequencing when results are unexpected or when key conclusions hinge on transcript architecture (below). In such contexts, the modest upfront effort is often outweighed by the substantial savings from avoiding costly, time-consuming artifacts.

“We strongly recommend that unbiased RNA sequencing should be used to assess the full repertoire of RNAs produced from key transgenes, particularly in light of unexpected or extraordinary results.”

Referee #3:

Plasmid based expression of transgenes is a foundational technology used in molecular biology, cell biology, genetics, genomics, and biotechnology. Included with this widespread use is the assumption that transfected cells express only the protein designed by the researcher. However, numerous studies have shown that transgenes often produce unexpected products. Despite being well-documented, the production of these unexpected products is almost entirely ignored by many researchers. This is a major problem for biomedical research, as much time, money, and effort is spent following false leads caused by transgene "misadventures". This manuscript is important in that it would highlight the issue for a wider audience of molecular biologists. It is, for the most part, a review of the numerous studies that have found such artifacts, with the addition of an example of splicing artifacts that can occur in a reporter library in Figure 3. Overall, I believe this would make a very nice review article that would be well-appreciated by the EMBO community of readers. I don't think there's enough new results if this was to be considered as a research article. I have a few comments that I believe should be addressed.

Author response: We thank the reviewer for their time, for their summary of our perspective, and for their insightful suggestions.

1. The authors should add a paragraph or two discussing circRNA plasmids that use inverted Alu repeats. Developed by Jeremy Wilusz's lab (PMID 25281217, 2014), this is designed to express circular RNA through "backsplicing" of a downstream 5' splice site to an upstream 3' splice site. This plasmid was used to claim that hundreds of mammalian genes have IRESes in 2021 (PMID 34437836) and to argue for IRES activity from the promoter regions of Hoxa9 and other genes this Spring (PMID: 40082722). However, the vast majority of RNAs expressed from this plasmid are linear and result from trans-splicing of 5' and 3' splice sites from separate mRNA molecules generated from internal promoters (PMIDs 33940551, 32955563, 33762403). Indeed, Dr. Wilusz wrote a 2021 review describing this artifact (PMID 33629517). This is an important additional example of another artifact type (trans-splicing) and highlights the fact that most researchers are not aware of these issues, even years after they are published.

Author response: This is an interesting topic that we overlooked, thank you for the feedback. We have added a section on highly synthetic systems which addresses this and your next comment, covering situations where researchers are engineering new, untested transcripts.

“In other cases, transgenes may be built from cDNA elements with significant synthetic contexts. ... As an example, circular RNA expression plasmids use flanking sequences with complementary inverted repeats to drive circularization of the RNA transcript by spliceosome-mediated back-splicing⁷⁶. While this system generates circular RNAs, it also produces abundant aberrantly spliced linear by-products⁷⁷, including trans-spliced RNAs arising from internal promoters and splicing between distinct molecules⁷⁸.”

2. An excellent example of splicing artifacts has been posted as a preprint in bioRxiv (PMID: 39149310). In this case, cryptic splicing was found to explain the results of multiple 3' UTR reporter libraries, including one authored by Sergej Djuranovic in 2018. Dr. Djuranovic has admirably posted a followup analysis showing these effects. This should be included in the manuscript.

Author response: Thank you for bringing this to our attention, and we have added the suggested references (including the full paper, published during our review process, PMID: 40715118):

“For instance, massive parallel reporter assays (MPRAs) place a library of candidate sequences into non-native surroundings to test for features like promoter activity, RNA stability, or translation efficiency (through interactions with transcription factors⁷³ and RNA-binding proteins⁷⁴). However, these highly synthetic architectures in MPRAs often unintentionally introduce RNA processing signals, confounding the readouts^{14,75}.”

3. The authors advocate requiring full-length plasmid sequence publication and RNA-seq validation of plasmids. I think this is a wonderful idea. However, wouldn't it be better for researchers to avoid these issues by using RNA transfections instead of plasmid transfections? Instead of finding out after the fact that an apparently "exciting" result was an artifact of cryptic promoters or splicing, losing several months of work along the way, direct RNA transfections would allow researchers to get more accurate results the first time around. It would also avoid temptation to malpractice.

Author response: We agree that RNA transfections are a useful orthogonal approach to avoid potential artifacts from DNA transfections, like spurious promoter use or unexpected RNA processing, and we have added RNA transfections as a suggested approach in the best practices section.

However, the use of *in vitro* transcribed RNA requires additional technical steps and is not free from its own issues, including potential dsRNA production and truncated transcripts (e.g., PMID: 39050733), so we are hesitant to suggest that it replace plasmid transfections entirely.

4. It would be fantastic for EMBO to lead the way in this by requiring authors to publish full plasmid sequences, and requiring (or at least strongly encouraging) RNA-seq validation of plasmids or RNA transfection validation, especially when plasmid transfection experiments are used to make extraordinary claims. They could also ask reviewers to consider whether plasmid transgenes could be making unintended products (linked to this and other reviews on the subject).

Author response: We, of course, whole-heartedly agree with this point! EMBO is well positioned to the lead. In light of referee 2's perspective that RNA-seq would be an additional burden on researchers, we suggest that a reasonable first step is to raise awareness, making researchers (including editors and reviewers) more aware of potential unintended RNA products produced in routine experiments involving transgenes.

Dear Dr. Jain,

I am pleased to inform you that your manuscript has been accepted for publication in the EMBO Journal.

You may qualify for financial assistance for your publication charges - either via a Springer Nature fully open access agreement or an EMBO initiative. Check your eligibility: <https://link.springer.com/journal/44318/how-to-publish-with-us>

Yours sincerely,

Cornelius Schneider, PhD
Editor
The EMBO Journal
c.schneider@embojournal.org

Please note that it is The EMBO Journal policy for the transcript of the editorial process (containing referee reports and your response letters) to be published as an online supplement to each paper. If you should prefer removal of any referee-only figures included in the point-by-point response(s), e.g. because they may still be used for future publication or because they have been reproduced from published work by others, please do let us know immediately via response email.

More information is available here: <https://link.springer.com/partners/embo-press/editorial-policies#Peer%20review>